

# The determinants and engagement patterns of chaperones and chauffeurs by Australian doctors in after-hours house-call services

Chris Onyebuchi Ifediora

School of Medicine, Griffith University, Gold Coast, Queensland, Australia

Corresponding author
Chris Onyebuchi Ifediora,
c.ifediora@griffith.edu.au

## ABSTRACT

**Objectives**. The use of escorts (chauffeurs and chaperones) while on duty in after-hours-house-call (AHHC) is one key protective option available to doctors in the service, and has been linked to low burnout and increased satisfaction in AHHC. This study aims to explore the patterns of engagement of escorts in Australian AHHC.

**Method**. This is a questionnaire-based, electronic survey of all 300 doctors involved in AHHC through the National Home Doctor Service (NHDS), Australia's largest providers of the service. The survey explored the doctor's experiences over the 12-month period from October 2013 to September 2014.

**Results**. This survey received a total of 168 valid responses, giving a response rate of 56%. Nearly 61% of the doctors involved in AHHC engaged escorts (chauffeurs and chaperones). Of those doctors that engage chauffeurs, three-quarters do so "all or most times", while only one-quarter engaged chaperones to the same degree of frequency. Hiring escorts is very popular among Brisbane (91.7%) and Sydney-based (88.2%) practitioners, but is unpopular in the City of Gold Coast (26.1%). There were moderate patronages in Adelaide (52.9%) and Melbourne Area (46.4%). Compared to males, females were less likely to drive themselves without escorts (OR 0.20; $P < 0.01$; CI [0.07–0.57]), but more likely to engage chauffeurs (OR 5.87; $P = 0.03$; CI [1.16–29.77]). Practitioners who were apprehensive were three times more likely to either engage escorts as chauffeurs (OR 3.10; $P = 0.04$; CI [1.05–9.15]) or as an accompanying chaperone if they self-drive (OR 3.03; $P = 0.02$; CI [1.16–7.89]).

**Conclusion**. More needs to be done to increase the engagement of escorts by doctors involved in the Australian AHHC, particularly given their proven benefits in the service. Future studies may be needed to fully explore the real reasons behind the significant associations identified in this study.

## INTRODUCTION AND BACKGROUND

Also known as Medical Deputizing Service (MDS), the after-hours house call (AHHC) service is becoming increasingly popular in Australia, with about 1.5 million patients benefitting from it in 2013 (*National Association for Medical Deputising Service, 2014*). The

same source posits that this number represents about 38% of all "urgent" after-hours presentations seen at the time, with Emergency Departments (EDs) accounting for another 62%. With this increase in demand comes the need to ensure that the welfare of the doctors providing AHHC services is not ignored in the quest for quality service-delivery. As a matter of fact, this welfare has come into focus lately following a recent finding that over half of the doctors involved in AHHCs have no protective measures in place while on duty, with the use of chaperones being one of the many protective options available to them (*Ifediora, 2015b*). The debate on this subject was further fueled by other online articles regarding use of chaperones and security measures in AHHCs (*Ozturk, 2015*; *Medical Observer, 2015*), while the perception persists that doctors involved in AHHC services face higher risks than their other colleagues (*Magin et al., 2005*). Even though no recent data exist on the pervading risks that Australian AHHC doctors face, a 2015 Germany study reported a 19% aggression rate over 12 months among a similar group of doctors (*Vorderwülbecke et al., 2015*).

These aforementioned discussions and concerns underline the need to evaluate the use of "escorts", a key aspect of security measures available to doctors involved in AHHCs. An escort is defined as "an attendant employed to accompany someone" (*Wordweb [Internet], 2015*), in this case, a doctor. In AHHCs, they are hired either as Chaperones or Chauffeurs. Basically, a chauffeur is "a person whose job is to drive people around in a car" (*Merriam-Webster, 2015*), while medical chaperones are "employed to accompany physicians during physical examinations, especially when the opposite gender is involved" (*Farlex, 2012*). The roles of escorts in AHHCs go beyond these primary roles, and may include working as security details, providing company for the doctor during drives between patients, carrying medical equipment, communicating on behalf of the doctor (like phone calls to patients or to the MDS office as needed), locating patients' addresses, and so on. Given that the use of these escorts is a key protective measure for doctors in AHHC services (*Ifediora, 2015b*), a knowledge of the doctors' attitudes on the engagement of these escorts is very important.

This study, therefore, aims to identify the patterns of use of escorts among Australian doctors involved in AHHCs. The independent doctor-variables that determine the use or otherwise of these escorts will also be explored. Not much exists in the literature regarding escorts in AHHC services, and answering these research questions will help reduce this existing knowledge-gap, and help policy makers and concerned doctors to consolidate, plan, improve and adjust their activities with respect to the welfare and safety of the practitioners in the industry.

Our findings may also have some international relevance, given that, as in 2011, 56% of Australian-based doctors were either born, or obtained their basic medical degrees, overseas (*Australian Bureau of Statistics, 2013*). It is expected that this cosmopolitan trend will continue, and results from our survey might help concerned practitioners and personnel involved in the international recruitment and movement of doctors make better-informed decisions. Furthermore, healthcare managers in a number of countries with similar services (like the United Kingdom, France, Canada, the Netherlands, among others), may find our results helpful as they design their own systems (*National Association for Medical Deputising Service, 2014*).

It should be noted that, in Australian AHHC, the choice of whether to employ an escort or not rests with the doctor, but some service providers do facilitate these engagements. Some doctors engage the escorts as chauffeurs (who, in some cases also act as chaperones), others drive themselves but still have them as chaperones, while some do not engage them at all, and prefer to drive and work alone. The escorts generally negotiate fees with the doctors that employ them, but our private enquiries reveal rates of between AU$20 to AU$40 per hour.

## METHODS

### Setting and participants

This study surveyed the patterns of engagement of escorts by the participants over a 12-month period spanning from October 2013 to September 2014. The Participants included all Australian-based medical practitioners (General Practitioners, GPs, and others) who undertake AHHCs through the National Home Doctor Service (NHDS), Australia's largest AHHC service provider (*National Home Doctor Service, 2014*). As at the time of this survey, the company rendered services in a number of locations, including Sydney, the Gold Coast, Adelaide, Brisbane Area (which also includes the Sunshine Coast), and Melbourne Area (which includes Geelong and Canberra) (*National Home Doctor Service, 2014*). The terms "Melbourne and Brisbane Areas" reflect the NHDS administrative groupings, and were not based on geographical or political classifications. Official NHDS sources indicated that there were 300 doctors working for them at the time of this survey, and this represents the study population as the study reached out to all of them. Each NHDS-location is overseen by a Clinical or General Manager, and, given that the company had successfully annexed the largest after-hours general practice clinics in most Australian major towns and cities over the few years prior to this study, it can be safely assumed that a study of NHDS-doctors reasonably represented the Australian after-hours doctor-population at the time.

The participants were contacted through e-mails sent directly to them by the managers overseeing their respective locations. A total of two reminders were sent out at fortnightly intervals after the initial dispatch. Data collection took approximately six weeks, from the end of September 2014 to the middle of November 2014.

### Questionnaire

The SurveyMonkey[R] software was used in the design and collation of the questionnaire, which was an 11-paged, electronic document divided into seven sections with a total of 25 questions designed to collect data for multiple studies (Data S1). The aspect relating to this survey covered Pages 1–4, with a total of 14 questions. As no validated, off-the-shelf questionnaire existed to answer the key research questions of this survey, a suitable tool was devised and its validity tested in a pilot survey of 10 Australian-based GPs who were not part of the study population. Recommendations and observations arising from the pilot study were used to modify the relevant sections of the draft questionnaire, culminating in the final tool.

## Analysis

Analysis was performed with *IBM SPSS Version 22*, aiming to answer the two key research questions of "patterns of engagement of escorts among doctors involved in AHHCs" and "associations between escort-engagements and various doctor-variables". For the analysis, a respondent is considered to have used an escort if, at any time in the period surveyed, the doctor had engaged the services of an escort for work in AHHCs. To give an idea of the frequency of usage, the responses included options on a 5-point Likert Scale, including, "not at all", "rarely", "sometimes", "most times" and "all the time". Where necessary, all responses of "not at all" is considered a "non-use" of escort services, while any engagement of escorts over the 12-months period, irrespective of the frequency, is categorized as "use". This dichotomy is justified based on the fact that doctors who engage chauffeurs and chaperones at all for their work in AHHCs are likely to share similar characteristics or ideologies (albeit to different degrees). Such traits are likely to differentiate them from doctors who do not engage these services at all.

The first research question identified the working habits of the doctors (driving self and working alone, driven by chauffeurs, and those that drive themselves but have an accompanying chaperone). Results for these are analyzed and stratified based on location of service by the doctors, and the "2-sided Fisher's Exact Test" was used to explore if there were significant differences between doctors who practice some habits and those who do not.

The second research question explored possible associations between the three driving habits (which form the dependent variables) and ten independent doctor-variables which were all presented as dichotomous variables (some were re-coded where necessary). These independent variables include gender (female and male), age (<40 years and ≥40 years), specialty (GPs and Non-GPs), postgraduate vocational status (fellowship attained and fellowship not attained), duration in after-hours service provision (≤2 years and >2 years), marriage status (in a legally-recognised social union or not), hours worked per week (<24 h and ≥24 h), living with children (yes and no), country of primary degree (Australian-trained and overseas-trained), and whether the doctor was apprehensive with the job (yes and no).

Given that the dependent variables were all "categorical data" presented in two categories as above, Binary Logistics Regression (BLR) conducted in two stages was used to identify significant associations. The first stage (Stage 1) was a univariate BLR analysis performed separately for each of the ten aforementioned independent variables. Only those found to be statistically significant were included in the final (Stage 2) multivariate BLR analysis. For each comparison, an odds ratio (OR) was presented, along with its corresponding 95% confidence interval (CI).

All significance levels (*p* values) are set at <0.05.

## Ethical considerations

Ethical clearance was obtained from the Human Research Ethics Committee of the Griffiths University, Australia (GU Ref No: MED/47/14/HREC), prior to commencing the study.

## RESULTS

### Basic demographics

A total of 168 valid responses were received out of the 300 questionnaires that were dispatched, giving a 56.0% response rate. The basic demographics of the respondents are summarized in Table 1.

### Patterns of escort engagements

As shown in Table 2, it was found that 96 out of 158 respondents (60.8%) that completed the relevant sections engaged an escort when working in AHHCs at any time during the 12 months surveyed. The table also reveals a wide variation in the level of engagements by location, with an overwhelming majority of the doctors working in Sydney (88.2%) and the Brisbane Area (91.7%) patronizing them, while only 26.1% did the same on the Gold Coast.

Table 3A further shows that 39.2% of the entire respondents used escorts all the time while at work for the period surveyed, while 26.2% did not use them at all at any time over this same period. When those involved in any of the three engagement habits to any degree are considered alone (i.e., excluding the "not-at-all" responses; Table 3B), it was found that about three-quarters of the doctors who engaged chauffeurs do so either "all the time" (53.1%), or "most times" (24.0%). A similar rate (73.1%) was found among those who drive themselves without an accompanying escort, comprising of 51.6% for those who drive alone "all the time" and another 21.5% who do so "most times". Conversely, among those who chose to drive themselves but still have an accompanying chaperone, only about a quarter (25.9%) do so at the same frequency as the other two groups (14.8% for "all the time" and 11.1% for "most times"). The significance of these are explored in the "Discussion" Section.

Stratification of the driving habits by location (Table 4) gives an insight into the patterns of all three driving habits. It was observed that there was a statistically significant difference between the doctors who "self-drive" to any degree, and those who do not do this at all, and that this difference applies in all the five locations identified (Table 4A). Similarly, among doctors who engage escorts to any degree (Table 4B) there were statistically significant differences between them and those who do not engage these services at all, and this is so in all five locations apart from the Melbourne Area (Fisher's Exact Test = 0.238). No such significance was found among doctors who self-drive but have accompanying escorts (Table 4C).

### Associations of escort-usage with independent doctor-variables

A number of significant predictors of driving habits were identified (Table 5). Firstly, it was found that female doctors were 80 times less likely to drive themselves without an accompanying escort (OR 0.20; $P < 0.01$; CI [0.07–0.57]), but nearly six times more likely to engage escorts (OR 5.87; $P = 0.03$; CI [1.16–29.77]). It was also observed that doctors who were apprehensive with the AHHC services were three times more likely to either engage escorts, either as chauffeurs (OR 3.10; $P = 0.04$; CI [1.05–9.15]) or as an accompanying chaperone while they self-drive (OR 3.03; $P = 0.02$; CI [1.16–7.89]).
**Table 1  Summary of the basic demographics of the respondents involved in Australian after-hours doctor house calls.**

| Statistic | Parameters | N | % |
|---|---|---|---|
| Gender<br>Valid = 168 | Male | 135 | 80.4 |
| | Female | 33 | 19.6 |
| Age range (Yrs)<br>Valid = 168 | 39 or less | 69 | 41.1 |
| | 40–60 | 90 | 53.6 |
| | Over 60 | 9 | 5.4 |
| Vocational/Registration status<br>Valid = 137 | Vocationally registered (Fellows) | 61 | 44.5 |
| | Non-vocationally registered (Non-fellows) | 76 | 55.5 |
| Primary degree<br>Valid = 160 | Australian-trained | 45 | 28.1 |
| | Overseas: New Zealand | 6 | 3.8 |
| | Overseas: other | 109 | 68.1 |
| Specialty<br>Valid = 160 | General practice | 135 | 84.4 |
| | Medical | 7 | 4.4 |
| | Surgical | 2 | 1.3 |
| | Emergency Department | 6 | 3.8 |
| | Other[a] | 10 | 6.3 |
| Location of service<br>Valid = 160 | Adelaide | 51 | 31.9 |
| | Brisbane area[b] | 36 | 22.6 |
| | Gold Coast | 23 | 14.4 |
| | Melbourne Area[c] | 31 | 19.4 |
| | Sydney | 17 | 10.6 |
| | Other (unfixed location) | 2 | 1.3 |
| Duration in after-hours<br>valid = 160 | <2 yrs | 80 | 50.0 |
| | ≥2–10 yrs | 54 | 33.8 |
| | >10 yrs | 26 | 16.3 |
| Hours worked/week<br>Valid=160 | <24 hrs/week | 62 | 38.8 |
| | 24–37.5 hrs/week | 47 | 29.4 |
| | >37.5 hrs/week | 51 | 31.9 |
| Marital status<br>Valid=168 | *Married* | 140 | 83.3 |
| | *Single* | 12 | 2.4 |
| | *De facto*[d] | 10 | 6.0 |
| | *Separated* | 4 | 2.4 |
| | *Widowed* | 2 | 1.2 |
| Whether protective<br>measures used or not<br>Valid = 151 | Yes | 65 | 43.0 |
| | No | 29 | 19.2 |
| | Have never thought about it | 9 | 6.0 |
| | Have thought about it, but unsure of what to do | 48 | 31.8 |

**Notes.**

[a]Others = Occupational physicians, Paediatricians, Public Health, etc.
[b]Brisbane Area = Brisbane and Sunshine Coast.
[c]Melbourne Area = Melbourne, Geelong and Canberra.
[d]De facto = Co-habitation and civil partnership.
**Table 2** Frequencies of escort-engagements by location among doctors involved in after-hours house calls.

| Location of service | Total respondents | Total number engaging escorts | Percentage use of escorts by location |
|---|---|---|---|
| Adelaide | 51 | 27 | 52.9 |
| Brisbane area | 36 | 33 | 91.7 |
| Gold coast | 23 | 6 | 26.1 |
| Melbourne area | 31 | 15 | 48.4 |
| Sydney | 17 | 15 | 88.2 |
| Totals | 158 | 96 (60.8%) | |

**Table 3** Patterns of escort-engagements among doctors involved in after-hours house calls.

| A. Overall engagement patterns | | | |
|---|---|---|---|
| Frequency of use | Self-drive Only (%) | Chauffeur-driven only (%) | Self-drive but accompanied by Chaperone (%) |
| Not at all | 38 (29.0) | 34 (26.2) | 61 (69.3) |
| Rarely | 8 (6.1) | 3 (2.3) | 7 (8.0) |
| Sometimes | 17 (13.0) | 19 (14.6) | 13 (14.8) |
| Most times | 20 (15.3) | 23 (17.7) | 3 (3.4) |
| All the time | 48 (36.6) | 51 (39.2) | 4 (4.5) |
| Totals | 131 (100.0) | 130 (100.0) | 88 (100.0) |

| B. Engagement frequencies for particular escort habits (excludes the "not at all" responses from Table 3A) | | | |
|---|---|---|---|
| Frequency of use | Self-drive only (%) | Chauffeur-driven (%) | Self-drive but accompanied by chaperone (%) |
| Rarely | 8 (8.6) | 3 (3.1) | 7 (25.9) |
| Sometimes | 17 (18.3) | 19 (19.8) | 13 (48.1) |
| Most times | 20 (21.5) | 23 (24.0) | 3 (11.1) |
| All the time | 48 (51.6) | 51 (53.1) | 4 (14.8) |
| Totals | 93 (100.0) | 96 (100.0) | 27 (99.9) |

No statistical associations were found between any of the escort-engagement behaviours of the doctors and their age, specialty, duration in AHHC, hours worked per week, marital status, postgraduate vocational status, and whether they live with children or not. The country where they obtained their primary medical degree also had no influence on their attitudes on whether to engage escorts or not.

## DISCUSSION

Male doctors comprised 80.4% of the respondents, while the remaining 19.6% were females, indicating that female doctors were represented less in AHHCs compared to their proportion in the Australian general practice population (where 43 per cent of doctors are females) (*Australian Bureau of Statistics, 2013*). This may be related to the higher apprehension female doctors express regarding the safety of AHHC, as was found

**Table 4 Summary of cross-tabulation results of the driving habits among doctors involved in after-hours house calls, by location.**

### A. Chauffeur-driven only

| Location of service (Total = 128) | Response (Chauffeur-driven or not) | Fischer Exact Test (2 sided) |
|---|---|---|
| Adelaide (n = 40) | Yes = 17<br>No = 23 | 0.021[*] |
| Brisbane area (n = 35) | Yes = 31<br>No = 4 | 0.000[*] |
| Gold coast (n = 16) | Yes = 1<br>No = 15 | 0.000[*] |
| Melbourne area (n = 22) | Yes = 10<br>No = 12 | 0.238 |
| Sydney (n = 15) | Yes = 15<br>No = 0 | 0.000[*] |

### B. Self-drive only

| Location of service (Total = 125) | Response (Self-driven or not) | Fischer Exact Test (2 sided) |
|---|---|---|
| Adelaide (n = 41) | Yes = 29<br>No = 12 | 0.013[*] |
| Brisbane area (n = 25) | Yes = 2<br>No = 23 | 0.000[*] |
| Gold coast (n = 23) | Yes = 21<br>No = 2 | 0.000[*] |
| Melbourne area (n = 28) | Yes = 15<br>No = 13 | 1.000[*] |
| Sydney (n = 8) | Yes = 1<br>No = 7 | 0.023[*] |

### C. Mixed driving habit (Self-drives, but chaperone present)

| Location of service (Total = 86) | Response (Mixed chauffeur and self) | Fischer Exact Test (2 sided) |
|---|---|---|
| Adelaide (n = 34) | Yes = 4<br>No = 30 | 0.427 |
| Brisbane area (n = 18) | Yes = 2<br>No = 16 | 0.633 |
| Gold coast (n = 11) | Yes = 0<br>No = 11 | 0.588 |
| Melbourne area (n = 18) | Yes = 0<br>No = 18 | 0.337 |
| Sydney (n = 5) | Yes = 1<br>No = 4 | 0.353 |

**Notes.**
*Statistically significant.
**Table 5** Binary Logistics Regression (BLR) showing the associations between the escort-engagement habits of doctors in after-hours call services and independent doctor-variables.

| Habit | STAGE 1 BLR | | | STAGE 2 BLR | | | | |
|---|---|---|---|---|---|---|---|---|
| | Independent doctor-variables | Odds ratio OR | Significance (*p*-value) | Independent doctor-variables | Odds ratio (OR) | Significance (*p*-value) | 95% CI of OR | |
| | | | | | | | Lower | Upper |
| Self-drive only (N = 127) | Gender (Female Vs Male) | 0.17 | 0.001 | Gender (Female Vs Male) | 0.20 | 0.002* | 0.07 | 0.57 |
| | Age range (Yrs) (≥40 Vs <40) | 0.41 | 0.04 | Age range (Yrs) (≥40 Vs <40) | 0.52 | 0.15 | 0.22 | 1.26 |
| Chauffeur-driven only (N = 110) | Gender (Female Vs Male) | 5.33 | 0.03 | Gender (Female Vs Male) | 5.87 | 0.03* | 1.16 | 29.77 |
| | Vocational status (VR[a] Vs Non VR[a]) | 3.43 | 0.01 | Vocational status (VR[a] Vs Non VR[a]) | 1.78 | 0.35 | 0.53 | 5.98 |
| | Primary degree (ATD[b] Vs OTD[c]) | 2.51 | 0.03 | Primary degree (ATD[b] Vs OTD[c]) | 1.99 | 0.21 | 0.67 | 5.91 |
| | Apprehension (Apprehensive Vs not apprehensive) | 3.20 | 0.02 | Apprehension (Apprehensive Vs not apprehensive) | 3.10 | **0.04*** | 1.05 | 9.15 |
| | Duration in Job (≤2yrs Vs >2 yrs) | 0.01 | 0.34 | Duration in Job (≤2 yrs Vs >2yrs) | 0.52 | 0.27 | 0.16 | 1.67 |
| Self-drives with chaperone (N = 84) | Apprehension (Apprehensive Vs not apprehensive) | 3.03 | 0.02 | Apprehension (Apprehensive Vs not apprehensive) | 3.03 | **0.02*** | 1.16 | 7.89 |

**Notes.**

All figures are shown to 2 decimal places, unless where $p < 0.01$, in which case it is show to 3 decimal places.

Most independent doctor-variables had no significant associations after Stage 1 of the BLR analysis and were not shown. They include age, specialty, duration in AHHCs, and country where primary degree was obtained.

All variables from the Stage 1 BLR were included for the Stage 2 analysis.

Confidence Intervals (CIs) were not shown in the Stage 1 BLR Column because they were not relevant; they were shown in the Stage 2 Column.

All the significant results are marked with asterisks (*).

[a]VR, Vocationally registered (attained postgraduate fellowship).

[b]ATD, Australian-trained Doctor.

[c]OTD, Overseas-trained Doctor.
in another study (*Tolhurst et al., 2003*). Interestingly, 71.9% of respondents were trained overseas, re-enforcing the international relevance of our study.

It is interesting to find that about three out of five doctors involved in AHHC services engaged an escort while on duty. Even though this represents a majority, it is not clear why the remaining two-fifths do not patronize escorts despite their proven benefits (*Ifediora, 2016*; *Ifediora, 2015a*). One line of thinking is that the fees involved may limit the patronage of these escorts as the doctors have to pay them from their own pockets. Another possibility is the need for privacy, as some doctors may feel that having these individuals as company may impact their privacies. Unfortunately, there is no existing study to allow a comparison to this finding, and our survey did not explore the real reasons behind the non-patronage for those who do not do so. It may be important to have future studies look into this, given that the use of chaperones in AHHCs (and possibly chauffeurs) have been identified as a protective measure in the service (*Ifediora, 2015b*), and that the availability of such protective measures are significantly linked to reduced burnout (*Ifediora, 2015a*) and increased satisfaction (*Ifediora, 2016*) in AHHC service deliveries.

The wide variation in patronage by location is another puzzling finding from this work. It is not clear why roughly 9-in-10 doctors working in Sydney and Brisbane engage escorts, while only about half do so in each of Adelaide and the Melbourne Area. Even worse is the Gold Coast, where only one-quarter engage them. There is a chance that there might be peculiarities in the characteristics of the doctors in these locations, whether in terms of ideological differences or in other respects. Again, future studies, perhaps with a qualitative approach, may be the best way to explore these findings.

Interestingly, there is a big difference in the degree to which doctors engage escorts, work alone or combine both (Table 3B). Nearly four out of five doctors who engage chaperones do so either "all the time" (53.1%) or "most of the time" (24.0%), and this shows that there is a strong commitment to the use of escorts among those who patronize them. Conversely, those who chose to drive themselves but have an accompanying chaperone, do so either rarely (25.9%) or only "sometimes" (48.1%). This may indicate that doctors who bother driving themselves generally prefer to go alone anyway, and would therefore have little need for chaperones most of the time. These are important findings for policy developments, as a knowledge of the real reasons for why doctors chose not to employ the services of chauffeurs would help in the campaign to get more of them interested in the service if these reasons are adequately tackled.

Finally, our observation that doctors that admitted to being apprehensive while on AHHC services were more likely to engage escorts (whether as chauffeurs or as accompanying chaperones) comes as no surprise. Those worried about their safety are more likely to adopt protective measures, and given that engagement of escorts is a proven safety measure among doctors engaged in AHHCs (*Ifediora, 2015b*), one would expect apprehensive doctors to engage them more.

## Study limitations

One major limitation is that this study was not designed to explore the real reasons behind the escort-engagement behaviours of the AHHC doctors. As indicated in the survey, future

surveys may do well to address this limitation. Another limitation is the non-representation of a few locations in Australia, specifically the towns and cities in States of Tasmania and Western Australia, as well as in the Northern Territory. Unfortunately, the AHHC services in these areas were few and not well developed at the time of this survey, and, as such, the study outcome is unlikely to have been significantly affected by this limitation. Finally, the small sample size is considered a limitation, but this is not an unusual limitation faced by studies of similar design, as online surveys are generally known to have poor response rates (*Penwarden, 2014*).

## Conclusions

This study concludes that 60.8% of doctors involved in after-hours house call services engaged escorts (chauffeurs or chaperones) while at work, leaving about 2-in-5 of the practitioners without this support. The differences among doctors who self-drive or engage chauffeurs across most Australian locations are statistically significant, and there was a wide variation by location in the escort-engagement behaviours, with major patronage in the Brisbane Area and Sydney, moderate patronage in Adelaide and Melbourne Area, and minimal patronage in Gold Coast.

Another conclusion is that doctors are very likely to be committed to their behaviours regarding the engagement of escorts or otherwise, since those who patronize them are likely to do so often, while those who prefer to work by themselves without escorts are very likely to only engage them sparingly.

Finally, the study concludes that female doctors are significantly less likely to work alone without escorts, but more likely to engage them. Also, practitioners that are apprehensive of their safety in AHHCs are more likely to engage the services of escorts.

## Recommendations

This study recommends that more needs to be done to encourage a higher number of doctors involved in AHHCs to engage escorts, particularly given the protective and psychological benefits they may offer these doctors. It is equally important that future studies fully explore the real reasons behind the patterns of escort-engagements displayed by doctors, as identified in this study.

### Funding

The author received no funding for this work.

### Competing Interests

The author declares there are no competing interests.

### Author Contributions

- Chris Onyebuchi Ifediora conceived and designed the experiments, performed the experiments, analyzed the data, contributed reagents/materials/analysis tools, wrote the paper, prepared figures and/or tables, reviewed drafts of the paper.

## Ethics

The following information was supplied relating to ethical approvals (i.e., approving body and any reference numbers):

Human Research Ethics Committee of the Griffiths University, Australia (GU Ref No: MED/47/14/HREC) approved this study.

## Data Availability

The raw data has been supplied as a Supplemental Information 1.

## Supplemental Information

Supplemental information for this article can be found online at http://dx.doi.org/10.7717/peerj.3218#supplemental-information.

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
