# Peer review of "The determinants and engagement patterns of chaperones and chauffeurs by Australian doctors in after-hours house-call services"

_PeerJ, doi:10.7717/peerj.3218_

## Round 0.1 · original submission · Major Revisions

· Academic Editor

Major Revisions

Please pay attention to comments of both reviewers.

·

Basic reporting

In this interesting paper the author is addressing the issue of the use of chaperones and chauffeurs by doctors' while performing their duties after-hours in the patient's home. Even though the research questions are worth investigating, the biggest flaw of the manuscript is data analysis and the presentation of the results. If the author appropriately amends these issues, I would only then recommend this manuscript for publication in PeerJ.

Minor comments:

1. The title suggests that this paper is investigating the issue of doctors’ safety, but the first (and last) mentioning of this word in the body of the manuscript is in the Discussion section (line 179), aside the key word “safety” which should be removed from this list because the paper is not actually dealing with the safety issues. In fact, there is no mention of the reasons why doctors engage or do not engage the services of escort, which could add valuable information to this work.
2. In the Introduction section there is no objective description of the actual need for an escort service, in example, how frequently were security issues recorded in the past and which kind of safety issues do doctors face in the investigated locations. Additionally, there was not one question from the survey presented here regarding the doctor’s personal attitude and perception regarding one’s personal safety during after-hours house call services in the areas they cover, which would (I suppose) propel the use of escort in cases when doctors do not feel safe.
3. No sound data is provided regarding representativeness of the sample and how many other agencies are there on the market and how many doctors use their services.
4. Lines 46/47 states that “about 1.5 million patients benefitted from them.“ without a reference, and whether there was that many after-hours house call (AHHC) services or it is the actual number of people who used those services, which is not the same.

Experimental design

1. Line 119-121: “a respondent is considered to have used an escort if, at any time in the 12-month survey period, the doctor had engaged the services of an escort for work in AHHC” – my opinion is that such a wide definition of escort usage (at least once in a year) is not appropriate because it does not describe the usual behavior. For example, if this was my work, I would classify the doctors who responded “most times” and “all the time” (line 122) as escort users, while those who responded “not at all”, “rarely” and “sometimes” I would regard as infrequent and never-users, which would then be binary variable and it would allow the use of logistic regression. The same procedure should be done for all 3 possibilities (“driving self and working alone”, “driven by chauffeurs”, and “drive themselves but have an accompanying chaperone”).

2. I did not see any mention of the potential combination of the 3 possibilities of „driving habits“, for instance one doctor could sometimes drive him/her-self and working alone, and sometimes use any of the other two options. How were those doctors classified/allocated to only one of the 3 groups?

Validity of the findings

1. The most serious issue is the approach used in data analysis, which in my opinion was not done properly when “a multi-staged Ordinal Logistics Regression (OLR)” (line 135) was used. My argument is that the dependent variable is not “ordinal categorical”, since there were 3 possible values of this variable „driving habits“ (driving self and working alone, driven by chauffeurs, and those that drive themselves but have an accompanying chaperone). This is a categorical variable because there is no particular ordering between 3 possible values in order to be named ordinal variable. So, in my opinion 3 separate models of logistic regression analysis should be formed (one for “driving self and working alone”, one for “driven by chauffeurs”, and one for “those that drive themselves but have an accompanying chaperone”), and in each model all 9 independent variables should be included.

2. In lines 123-125 the research question is named, but no statistical testing was performed to provide the answer to “The first research question identified the working habits of the doctors (driving self and working alone, driven by chauffeurs, and those that drive themselves but have an accompanying chaperone) and their frequencies”. Also, Table 4 has no added information above the information already provided in Table 3. I would suggest the provision of frequency of escort use according to the location of service, by joining Table 2 and Table 3 in one table. And at least a chi-square test (or Fisher exact test, depending on expected frequencies) should be done here to show if there is any statistically significant difference in the frequency of escort use regarding the location.

Additional comments

I

·

Basic reporting

Thank you for investigating this important and neglected aspect of AHHC services. You’ve obviously given it a lot of thought and are concerned about the safety and well-being of colleagues and how this can be improved. Your hard work has obviously paid off, given the number of publications that have arisen from this study. Congratulations!

For this paper to be published there are a number of issues that need to be addressed to make it more robust and easier to understand. Please address the following:

1. Basic reporting
Line 47: This figure would benefit from a reference.
Line 48: There is a grammatical error in this sentence: ˝welfare˝ is singular; therefore, it should state ˝the welfare of doctors is not ignored...˝.
Line 63: In this sentence ˝equipments˝ does not need an ˝s˝ to be plural. Your use of the word ˝communications˝ here is unclear-please explain in the text what you mean.
Line 64: The second ˝a˝ in this sentence is unnecessary. This sounds like an important point so I suggest explaining it in a bit more detail for those readers who are not familiar with you previous research.
Line 65: In this sentence ˝a knowledge of the doctors’ habits regarding their engagement˝ is confusing-presumably you’re referring to the escort’s engagement but it sounds like you’re referring to the doctors-please fix.
Line 71: There seems to be a word missing in this sentence. I would suggest adding ˝policies˝ after ˝as necessary˝ to make it clearer what you’re referring to.
Line 74-‘5: What relevance does this fact have to the topic studied? Are you reflecting upon earlier results? If so, please explain in the text.
Line 85: There seems to be a category missing here: what about doctors who employ chauffeurs who also serve as chaperones?
Line 106: In this sentence ˝despatch˝ has been misspelt-please fix.
Line 113: This line is a bit confusing the way it’s written. I suggest the following alternative: ˝...in this study, a tool was devised and its validity tested in a pilot survey of 10 Australian GPs who...˝.
Line 132: Although the use of the word ˝kids˝ is common, it is not appropriate for a professional scientific journal. Please replace with ˝children˝.
Line 146: It would be more appropriate to state ˝during the period under survey˝ instead of ˝at the period under survey˝-suggest replacing.
Line 152: Even though there is no word limit, readers’ time is limited so please don’t repeat phrases, such as ˝over the 12 month period˝ at the end of this sentence.
Line 158: Please correct spelling of chaperone.
Line 163: The significant variables do not need to be capitalised.
Lines 166-9: A sentence should not stand alone. A paragraph should preferably have at least three sentences.
Line 171: Please see above in relation to use of the word ˝kids˝.
Line 180-1: In this sentence ˝respondents˝ is plural so you don’t need to use ˝the˝; ˝to this work˝ is also superfluous, so please delete.
Line 197: You need to add ˝area˝ to Melbourne or delete the preceding ˝the˝ for this sentence to make sense.
Line 202: ˝Very˝ is a bit too much. ˝Interestingly˝ will suffice.
Line 206: In this sentence ˝chaperon˝ is misspelt again-please fix.
Line 224: Please fix spelling mistake.
References: Names of journals need to be capitalised, e.g., Australasian Medical Journal.
Line 25-6: It would be useful to add some dates so readers know when the study was conducted.
Line 27: It would sound better if you wrote ˝giving a response of 56%˝. It is not customary to start a sentence with a percentage; therefore, I suggest you write 61% in words.
Line 36: This sentence is confusing; weren’t the first more likely to work with and the latter without escorts?

Experimental design

The research question needs to be better defined. You state that your aim is to identify the frequency of escort use but you’ve already mentioned in the first paragraph of the Introduction that one third of doctors involved in AHHC employ chaperones. Please focus on what is new in this study. The research questions need to be relevant and meaningful.

Since you have provided very little information about the questionnaire in the Methods’ section, it would be great if you could add it to the Supplementary Materials.

Line 103-7: It would be helpful if you could clearly state the study design at the beginning of this section. As written, it gives the incorrect impression of being a cohort study. The last sentence in this section should come earlier.

An Ethics statement has not been provided in the Methods’ section of the manuscript. Please add.

Line 120: you mention a 12 month study period which is incompatible with a cross-sectional study design. Presumably you meant that the questions in the survey referred to the 12 month period prior to survey completion, i.e., to the engagement of escorts during the preceding 12 month period. If so, please correct.

Validity of the findings

In your tables you need to be consistent with your use of capitals in the headings and sub-headings; either capitalise each word or the first letter of the first word only.

Table 1: There is a missing larger-than sign for the variable duration of after-hours work 2-10 years. The percentage of single doctors is incorrect-please fix.

Table 2: The % for the Melbourne area is incorrect-please fix.

Line 150-2: This section is a bit confusing. In the text you refer to doctors who ˝used escorts all the time˝ but in Table 3, the corresponding percentage refers to doctors who were ˝chauffeur driven only˝ all the time. You’ve done the same with the other figure given. Based on the definition in the Intro, escorts include both chauffeurs and chaperones; therefore, the number of doctors who used escorts all the time should be the sum of the last two columns in Table 3. If this is not what you intended to show, then you need to be more precise as to what you are referring to in the text- please fix.

Table 3: The percentages in the ˝self-drive˝ column have been incorrectly calculated-please recalculate.

It seems as if there are other categories of escort-engagement missing from Table 3. What about chauffeur-driven and chaperoned?

I find Table 4 confusing because it is meant to refer only to doctors who engage escorts and yet in the first column you have included doctors who ˝self-drive only all the time˝, meaning, from my understanding, that they do not engage escorts, i.e., neither chauffeurs nor chaperones, but drive themselves and work alone. Please clarify.

Table 5: The title has redundant ˝the˝ in a few places. The ˝p˝ in p values should be written in italics. The first footnote is grammatically incorrect. VR stands for ˝Vocationally Registered˝ not ˝Vocationally Trained˝. Please fix.

Limitations: the small sample size, i.e., limited response is also a limitation of this study.

Recommendation: I suggest you soften the first sentence by adding ˝may˝ before ˝offer these doctors˝.

Additional comments

A lot of the results from the tables are repeated in the text, including the discussion and conclusion sections. I’m not aware of a word limit for this journal but if there is one, then I suggest cutting out unnecessary repetition of results.

Numbers, in brackets, should accompany percentages in the text.

---

## Round 0.2 · Minor Revisions

· Academic Editor

Minor Revisions

Thank you for addressing the reviewers' comments. There are some small suggestions from one of the reviewers and it would be good to address them.

·

Basic reporting

Thank you for working on improving the manuscript. There are still a few minor details that need to be fixed to aid comprehension and flow:

Line 46: the second ˝service˝ should be singular
Line 50: you don’t need ˝the˝ before ˝Emergency Departments˝, so please delete; I suggest replacing ˝popularity˝ with ˝demand˝
Line 51: I suggest replacing ˝doctors in AHHC˝ with ˝doctors providing AHHC services˝; if you use AHHC on its own then it should be plural, i.e, AHHCs, otherwise it should be followed with ˝services˝. Please correct throughout.
Line 58: ˝Data˝ is plural therefore it should be followed by ˝exist˝.
Line 58-9: this sentence is very confusing; I suggest you simplify it to ˝...risks Australian AHHC doctors face, a...˝
Line 63: ˝as˝ is unnecessarily repeated-please delete
Table 1: under ˝specialty˝, ˝Others˝ should be singular; under ˝Duration in after-hours˝, the second category should not overlap with the first
Line 172: there is a typo
Line 179: please delete the first ˝are˝-it creates confusion
Table 3A-please add bracket and remove unnecessary percentage sign
Line 200: ˝found that˝ is repeated unnecessarily
Line 230: there is an unnecessary ˝have˝
Line 251-7: this paragraph seems like an unnecessary repetition of the first paragraph of the Discussion- I suggest deleting
Line 286: typo-capital letter missing

Experimental design

Thank you for addressing my earlier comments.

Validity of the findings

Thank you for addressing my earlier comments.

---

## Round 0.3 · accepted · Accept

· Academic Editor

Accept

Thank you for addressing all comments of the reviewers.